# Interleukin 26 Induces Macrophage IL-9 Expression in Rheumatoid Arthritis

**DOI:** 10.3390/ijms24087526

**Published:** 2023-04-19

**Authors:** Yi-Hsun Wang, Yi-Jen Peng, Feng-Cheng Liu, Gu-Jiun Lin, Shing-Hwa Huang, Huey-Kang Sytwu, Chia-Pi Cheng

**Affiliations:** 1Graduate Institute of Life Sciences, National Defense Medical Center, Taipei 11490, Taiwan; 2Department of Pathology, Tri-Service General Hospital, National Defense Medical Center, Taipei 11490, Taiwan; 3Division of Rheumatology/Immunology and Allergy, Department of Medicine, Tri-Service General Hospital, National Defense Medical Center, Taipei 11490, Taiwan; 4Department and Graduate Institute of Biology and Anatomy, National Defense Medical Center, Taipei 11490, Taiwan; 5Division of Breast Surgery, Department of Surgery, New Taipei City Hospital, New Taipei City 241204, Taiwan; 6National Institute of Infectious Diseases and Vaccinology, National Health Research Institutes, Zhunan 35053, Taiwan; 7Department of Microbiology and Immunology, National Defense Medical Center, Taipei 11490, Taiwan

**Keywords:** rheumatoid arthritis, IL-26, IL-9, IRF4, AKT, FoxO1

## Abstract

Rheumatoid arthritis (RA) is an autoimmune disease with chronic inflammation, bone erosion, and joint deformation. Synovial tissue in RA patients is full of proinflammatory cytokines and infiltrated immune cells, such as T help (Th) 9, Th17, macrophages, and osteoclasts. Recent reports emphasized a new member of the interleukin (IL)-10 family, IL-26, an inducer of IL-17A that is overexpressed in RA patients. Our previous works found that IL-26 inhibits osteoclastogenesis and conducts monocyte differentiation toward M1 macrophages. In this study, we aimed to clarify the effect of IL-26 on macrophages linking to Th9 and Th17 in IL-9 and IL-17 expression and downstream signal transduction. Murine and human macrophage cell lines and primary culture cells were used and stimulated by IL26. Cytokines expressions were evaluated by flow cytometry. Signal transduction and transcription factors expression were detected by Western blot and real time-PCR. Our results show that IL-26 and IL-9 colocalized in macrophage in RA synovium. IL-26 directly induces macrophage inflammatory cytokines IL-9 and IL-17A expression. IL-26 increases the IL-9 and IL-17A upstream mechanisms IRF4 and RelB expression. Moreover, the AKT-FoxO1 pathway is also activated by IL-26 in IL-9 and IL-17A expressing macrophage. Blockage of AKT phosphorylation enhances IL-26 stimulating IL-9-producing macrophage cells. In conclusion, our results support that IL-26 promotes IL-9- and IL-17-expressing macrophage and might initiate IL-9- and IL-17-related adaptive immunity in rheumatoid arthritis. Targeting IL-26 may a potential therapeutic strategy for rheumatoid arthritis or other IL-9 plus IL-17 dominant diseases.

## 1. Introduction

Rheumatoid arthritis (RA) is a chronic inflammatory autoimmune disease with clinical signs including chronic inflammation of synovial joints, redness, and progressive cartilage and bone erosion [1,2]. The joint cavity of RA patients was infiltrated by many immune cells, including T help (Th) 9, Th17, macrophages, and osteoclasts [3,4]. Proinflammatory cytokine secretion by inflammatory immune cells and synoviocyte, including interleukin (IL)-1β, 6, 9, and 17, has been demonstrated to influence this disease [5,6,7,8,9]. Review of the literature showed that IL-26 was significantly detected in synovial fluid and serum of RA patients [10,11]. Previous studies reported that IL-26 was found to associate with many proinflammatory cytokines, such as IL-6, IL-10, IL-17A, TNF-α, and mediated recruitment of T help cells [10,12,13]. Besides RA, recent studies reported that IL-26 was also found in many inflammatory lung diseases, such as asthma and chronic obstructive pulmonary disease (COPD) [14]. Even in the recent COVID-19 pandemic, IL-26 was regard as a marker of acute hyperinflammation [15]. These inflammatory lung diseases were also linked to the IL-9 or Th9 as an allergic situation [16].

IL-26, a member of the IL-10 cytokine family, was first found in hybridizing Herpesvirus saimiri (HVS)-transformed T cells in vitro [17]. IL-26 transmitted signals through receptors IL-10RB and IL-20RA and further activated STAT1/3 related signals in colorectal carcinoma cell lines [18,19]. Although mice do not natively express the IL-26 protein, past studies confirmed that mice conserved with IL-26-related receptors IL-20RA and IL-10RB showed a similar progress to the human system [20,21]. Our previous studies reported that IL-26 could inhibit osteoclastogenesis and modulate monocyte differentiation toward proinflammatory M1-like macrophages with secreting ability of proinflammatory cytokines such as IL-6 and TNF-α [22,23]. Moreover, recent studies reported that IL-9 and IL-17A may come from many cell sources, such as macrophages and T helper cells [24,25,26]. However, there is no study reporting a relationship between IL-26 and IL-9 in macrophages in arthritic disease. Therefore, our purpose is to investigate whether IL-26 is involved in the regulation of IL-9 and IL-17 expression in rheumatoid arthritis. Understanding the functionality and recruitment capacity of IL-26 between IL-9 and IL-17 in macrophages may clarify a central role in RA or chronic inflammatory disease.

## 2. Results

### 2.1. IL-26^+^ and IL-9^+^ Macrophages Were Highly Expressed in Synovial Tissue in RA

Firstly, to clarify our hypothesis that IL-26 was involved in IL-9 production in macrophages of rheumatoid arthritis, synovial tissue from RA patients was collected and stained by hematoxylin and eosin (H&E) for identification of leukocyte infiltration and stained by immunofluorescence for checking the location of IL-26 and IL-9. Our results showed that the infiltrated synovium were highly expressing IL-26 and IL-9 cytokines (Figure 1). In addition, both cytokines were overlapped with CD68^+^ stained cells, respectively (Figure 1A). Moreover, counterstaining of IL-26 and IL-9 showed colocalization in synovial tissue of RA (Figure 1B).

### 2.2. Effect of IL-26 on IL-9 Cytokines Expression and CD80 Macrophage Differentiation in RAW264.7

Previous works reported that IL-26 is a strong inducer of IL-17, but none link it to IL-9. We wanted to test our hypothesis and obtain clear images showing that IL-9 can be directly stimulated by IL-26 in macrophages. We took advantage of murine cells without IL-26 and treated cells with human IL-26 recombinant protein, IL-4, or IFNγ, then analyzed them with CD80 cell surface marker and IL-9 and IL-17A cytokines, which are dominantly expressed in RA patients. Our results showed that IL-26 promoted murine macrophage cells toward CD80^+^ cell differentiation and upregulated the expression of IL-9 and IL-17A (Figure 2), which were previously regarded as the only dominant expression on Th9 and Th17.

### 2.3. Effect of IL-26 on IL-9, IL-17A-Related Transcription Factor Expression in RAW264.7

Previous studies reported that IRF4, RelB, PU.1, and RORγt were critical factors for Th9 and Th17 cell differentiation [27,28]. To test which factor was major involved in macrophage IL-9 and IL-17A expression, we treated RAW264.7 cells as per the description above then analyzed the messenger RNA gene expression level of PU.1, RelB, IRF4, and RORγt by quantitative PCR. Our results showed that IL-26 strongly increased the gene expression of IRF4 and mildly increased it in RelB, but not significantly in PU.1 and RORγt (Figure 3A). To further check if the gene expression was reflected to protein expression, cells were incubated for 6 to 12 h with IL-26, then protein was isolated and detected by Western blot. Our results showed that protein levels of IRF4 and RelB were also significantly increased. This result was consistent with mRNA gene expression pattern (Figure 3B,C).

### 2.4. Effect of IL-26 on AKT and FoxO1 Activation in RAW264.7 and THP-1

Sakshi et al. reported that PI3K/AKT was an upstream kinase of FoxO1 and modulated FoxO1 localization and transcriptional regulation in Th9 and Th17 [29]. In addition, Eleni et al. showed that the AKT signaling pathway plays an important role in the differentiation of M1/M2 macrophages and regulation of cytokine expression, activity, and apoptosis [30]. Hence, we test whether the AKT signaling pathways were involved in activation of macrophage after IL-26 stimulation in RAW264.7 cells. Our data showed that both AKT and FoxO1 phosphorylation were significantly increased after IL-26 stimulation (Figure 4A,B). To further confirm that the FoxO1 was regulated by AKT, we incubated RAW264.7 and THP-1 cells with MK2206, an AKT phosphorylation inhibitor accompanied by IL-26. Our results showed that the phosphorylated AKT and FoxO1 elevated by IL-26 were significantly eliminated by pretreating MK2206 (Figure 4C–F).

### 2.5. Effect of IL-26 plus MK2206 on IL-9- and IL17A-Regulated Transcriptional Gene Expression in RAW264.7 and THP-1

Moreover, to further clarify the effect of AKT-FoxO1 signaling in IL-26 stimulated IL-9- and IL-17A-expressing macrophages, we treated RAW264.7 and THP-1 cells with IL-26 and MK2206 alone or combined together, then analyzed the IL-9 and IL-17A upstream transcriptional gene regulation by qPCR and Western blot. Our results showed that IL-26 significantly increased IRF4 and RelB expression both in mRNA (Figure 5A,B) and protein levels (Figure 5C–F), and these effects were not reversed by MK2206. Interestingly, IRF4 expression was dramatically elevated by IL-26 plus MK2206 in human macrophage THP-1 cells.

### 2.6. Effects of IL-26 plus MK2206 on IL-9 and IL-17A Cytokines Expression in RAW264.7, THP-1, and Primary Cells

To further confirm phenotype effect of the AKT-FoxO1 signaling pathway on macrophage differentiation and cytokines expression in RAW264.7 and THP-1, we treated RAW264.7 and THP-1 cells with IL-26, MK2206, or IL-26 plus MK2206 then analyzed cells with CD80 and IL-9, IL-17A marker by flow cytometry. Our data showed that IL-26 upregulated the expression of IL-9 and IL-17A and promoted macrophage cells toward CD80^+^ cell differentiation. The phenomenon was inhibited by MK2206 both in RAW264.7 and THP-1 on IL-17A. Interestingly, IL-26 plus MK2206 synergistically promoted the macrophage cells expressing IL-9 in both cell lines RAW264.7 and THP-1 (Figure 6). To make sure the phenomenon was not a result of immortalized cells, primary cells from murine bone marrow-derived macrophages (BMDM) and human peripheral blood mononuclear cells (PBMC) were subjected to IL-26 stimulation. Similarly, our results showed that IL-26 significantly upregulated the expression of IL-9 in murine BMDM and human PBMC (Figure 7). These results showed that IL-26 may be involved in initiating IL-9 expression and have a synergistic effect with MK2206 in the AKT-FoxO1 pathway.

## 3. Discussion

In this study, we are the first group to find that IL-26 can directly promote the IL-9 expression and is involved in modulating IL-17A in macrophages. Over the most recent two decades in RA study, Susumu et al. showed IL-17-deficient mice with suppression effects in collagen-induced arthritis [31]. Francesco et al. found potential involvement of IL-9 and Th9 cells in the pathogenesis of RA [32]. All these studies pointed out that IL-9 represented by Th9 cells and IL-17A represented by Th17 cells were well known and mostly focused on the T cell pathogenesis in RA [6,8,33]. Recently, a new concept was established in which Th9 and Th17 are controversial twins in cancer immunity [34,35]. Corvaisier et al. were the pioneers linking IL-26 to macrophage and Th17 generation in RA [10]. However, no one discussed the relationship between IL-26 and IL-9 in RA. Based on our previous works, IL-26 played a multifunction role in macrophages, including M1 macrophage polarization and inhibiting osteoclastogenesis. Moreover, macrophages presented an emerging and local bridge role in immunoreaction. Thus, how IL-26 behaved in the regulation of IL-9 and Th9 and was involved in the IL-9/IL-17A axis became a very interesting point in RA. Here, our results show that macrophages express not only IL-17A but also IL-9 after IL-26 stimulation, which may initiate and facilitate Th9 as well as Th17 during T cell maturation.

The heterogenous macrophages and cytokine network in the progression of RA is complicated. Conventionally, the imbalance of M1 and M2 macrophages was regarded as a mediator to Th1 and Th2 in RA pathophysiology [36]. However, more and more subtypes of macrophage and cytokines had been identified as involved in this vicious circle. Nowadays, IL-9 and IL-17A are a new direction in rheumatic research. Transcriptional regulation of IL-9 and IL-17A was well documented in Th9 and Th17 differentiation. IRF4, RelB, and PU.1 played a critical role in the maturation and function of Th9 [27,37], and RORγt was the specific transcriptional regulator in Th17 [28,38]. Moreover, IRF4 also regulated IL-17A promoter activity and controlled RORγt-dependent Th17 cells [39]. RelB deficiency regulated expression of RORγt and RORα4 in thymic γδ T cell and downstream IL-17 [40]. In our study, we found that IL-26-stimulated macrophages showed dominant transcriptional regulation on IRF4 and RelB, not RORγt. This means that IL-26-stimulated macrophages are probably more specifically modulating on IL-9 than IL-17A. Interestingly, IRF4 was identified to be involved in the regulation of many immune cells, including, Th9, Th17, B cells, and M2 macrophages. This finding supported our previous work stating that IL-26-polarized CD80^+^ macrophages also produced IL-10 cytokine [23].

Focusing on the maturation of Th9 and Th17, previous studies reported that FoxO1 was a master regulator in balance of Th9 and Th17 [29,41]. Saksh et al. showed that FoxO1 is essential for binding and transactivating IL-9 and IRF4 promoters in Th9, Th17, and iTreg cells [29]. FoxO1 phosphorylation or restriction in cytosol would promote Th17 differentiation as well as IL-17A expression [42]. PI3K/AKT, an upstream kinase of FoxO,1 modulated FoxO1 localization and transcriptional regulation in Th9 and Th17 [29]. PI3K/AKT inhibition significantly enhanced IL-9 expression in Th9 cells. However, the role of FoxO1 in macrophage polarization was controversial and depended upon each condition. Yang et al. showed that FoxO1 was regarded as an M1 macrophage activator when its deficiency would change their gene expression pattern towards M2 phenotype [43]. Sangwoon et al. reported that FoxO1 is highly expressed in M-CSF-derived (M2-like) macrophages and preferentially enriched in the IL-10 promoter [44]. Although the full function of FoxO1 in macrophage was unclear, the regulation mechanism of AKT inactive FoxO1 was consistent with Th cells [45]. In our study, we found that IL-26 was a new regulator on IL-9 and IL-17A expression via AKT-FoxO1 signaling in both mouse and human macrophage cells. Furthermore, blockage of the AKT-FoxO1 pathway by the AKT inhibitor in IL-26 stimulation showed increased IL-9 expression in the macrophage population.

In conclusion, and summarized as Appendix A, we provided evidence that IL-26 stimulation is directly involved in IL-9 and IL-17A proinflammatory cytokine expression as well as CD80 M1 macrophage differentiation. We also found that IL-26 induced IL-9- and IL-17A-expressing macrophages via IRF4, AKT, and FoxO1 signaling pathways. Modulation of the mechanisms of AKT-FoxO1 by AKT inhibitor in IL-26-stimulated macrophages dominantly increased the ratio of IL-9-expressing macrophages. Taken together, our findings decipher the new role of IL-26 in macrophage subtype and proinflammatory cytokine regulation in innate immune systems. Targeting IL-26 might be a new strategy for IL-9-dominant inflammatory diseases, such as RA.

## 4. Materials and Methods

### 4.1. Reagents

Human IL-26 (MBS1362134) recombinant protein was purchased from MyBiosource (San Diego, USA). Human or mouse IL-4 (200-04, 214-14), IFN-γ (300-02, 315-05), and M-CSF (300-25, 315-02) recombinants were purchased from Peprotech (London, UK). Akt Inhibitor MK2206 (11593) was purchased from Cayman (Ann Arbor, MI, USA). STAT inhibitor, Nifuroxazide (46494-100MG), and all other reagents were purchased from Sigma-Aldrich (St. Louis, MO, USA).

### 4.2. Hematoxylin and Eosin (H&E) and Immunofluorescence (IF) Staining

Synovial tissue from RA patients was collected from Tri-Service General Hospital (Taipei, Taiwan; TSGHIRB approval no. C202205045 and 1-102-05-091). For hematoxylin and eosin staining, 5 μm paraffin-embedded tissue sections were deparaffinized and stained with hematoxylin for 40 s and with eosin (ab245880, Abcam, Cambridge, UK) for 30 s. For immunofluorescence staining, paraffin sections were incubated in the oven at 65 degrees for 2 h and then incubated with Trilogy reagent (920P, Sigma-Aldrich) for 1.5 h. After washing for 2 min twice with PBS, the tissue was permeabilized with 0.3% Triton X-100 for 10 min and blocked with PBS containing 5% FBS for 1 h. After the blocking procedure, the tissue was stained with primary antibodies (Appendix A) overnight at 4 °C, then washed for 10 min twice with PBS containing 5% FBS, stained with secondary antibodies (Appendix A) for 1 h, then washed for 10 min twice with PBS and mounted by inmounting medium (ab104139, Abcam). Samples were analyzed using an Olympus fluorescence microscope.

### 4.3. Cell Lines and Primary Culture

Murine RAW 264.7 and human THP-1 monocyte cell lines were purchased from the Food Industry Research and Development Institute (Hsinchu, Taiwan). Murine RAW 264.7 was cultured in Dulbecco’s Modified Eagle’s Medium (DMEM) (11995, Gibco, Waltham, MA, USA) containing 10% fetal bovine serum (FBS) (PS-FB2, Peak Serum, Wellington, CO, USA) and 1% penicillin–streptomycin (P/S) (03-031-1B, Biological Industries, Israel). Human THP-1 monocyte was cultured in Roswell Park Memorial Institute (RPMI) 1640 (22400, Gibco) containing 10% FBS, 1% P/S, and 55 nM 2-Mercaptoethanol (21985, Gibco).

Mice tibia and femur from four DBA-1/J mice was used for isolated primary murine bone marrow-derived macrophage (BMDM), then bone marrow flushing fluid was collected and filtered with 35 µm nylon mesh and cultured in 12-well plates with DMEM containing 10% FBS and 1% P/S. After 1 day of incubation, cells were washed with PBS and suspended cells were collected then treated with 50 ng/mL murine M-CSF (315-02, Peprotech, London, UK) for 1 day before further experiment.

Human whole blood samples from four healthy volunteers were used for isolated primary human peripheral blood mononuclear cells (PBMCs). PBMCs were isolated from fresh blood samples by density-gradient centrifugation with BioWhittaker™ Lymphocyte Separation Medium (17-829E, Lonza, Basel, Switzerland), then cultured in a 10 cm dish with RPMI 1640 containing 10% FBS, 1% P/S, and 50 ng/mL human M-CSF (300-25, Peprotech) for 2 days, changing the medium every day before further experiment.

### 4.4. Flow Cytometry

After incubation with different conditions, cells were harvested and washed twice with flow-staining buffer then stained with surface marker antibody (Appendix A) for 60 min. For intracellular staining, cells were washed twice and incubated with fixation buffer for 30 min then washed twice with 1X permeabilizing buffer (420801, 421002, Biolegend, San Diego, CA, USA) before antibodies conjugation. Cells were resuspended in 100 μL 1X permeabilizing buffer and stained with intracellular antibodies (Appendix A) for 60 min then washed twice by flow-staining buffer. Cells were analyzed with an Attune NxT V6 flow cytometer (Thermo Fisher Scientific, Waltham, MA, USA).

### 4.5. Protein Extraction

After incubation with different conditions, for whole cell protein extraction, cells were harvested and lysed with Mammalian Protein Extraction Buffer (28941279, GE Healthcare, Chicago, IL, USA) containing 1X Halt Protease and Phosphatase Inhibitor Cocktail (78442, Thermo Fisher Scientific, Waltham, MA, USA), then centrifuged at 1200× *g* for 20 min to collect the supernatant.

### 4.6. Western Blot

Protein extraction samples were run on 10.0% SDS-PAGE using the Mini-PROTEAN Tetra cell system (BioRad, Hercules, CA, USA) then transferred onto an Immobilon-P Membrane (IPVH85R, Merck Millipore, Burlington, MA, USA) in the same system. Membranes were blocked with TBS-T (T9142, Takarabio, Japan) containing 5% skimmed milk powder and stained with primary antibodies (Appendix A) overnight at 4 °C. Membranes were washed for 10 min twice with TBS-T and stained with secondary antibodies (Appendix A) for 30 min then washed for 15 min twice with TBS-T. Membranes were analyzed using UVP ChemStudio PLUS (Analytik Jena, Jena, Germany).

### 4.7. RNA Extraction and Real-Time PCR

After incubation with different conditions, RNA was extracted using the NucleoSpin^®^ RNA kit (MACHEREY-NAGEL, Germany) then underwent reverse transcription to cDNA using the SensiFAST™ cDNA Synthesis Kit (BIO-65053, Bioline, UK). Real-time PCR was analyzed on a LightCycler 96 system (Roche, Basel, Switzerland) with specific primers (Appendix A). The data were exported to the Lightcycler96 software, the Rel Quant mode was used, and the relative levels of each value were evaluated and normalized with GAPDH then calibrated to N groups. The quantitative thermal cycling parameters were 95 °C for 15 min, then 40 cycles for 30 s at 95 °C, 30 s at 60 °C, and 1 min at 72 °C, followed by extension for 10 min at 72 °C.

### 4.8. Statistical Analysis

Data are displayed as means ± SD and were analyzed by using one-way ANOVA, followed by Tukey multiple comparisons in post-test. *p* < 0.05 was considered as statistically significant.

## Figures and Tables

**Figure 1 ijms-24-07526-f001:**
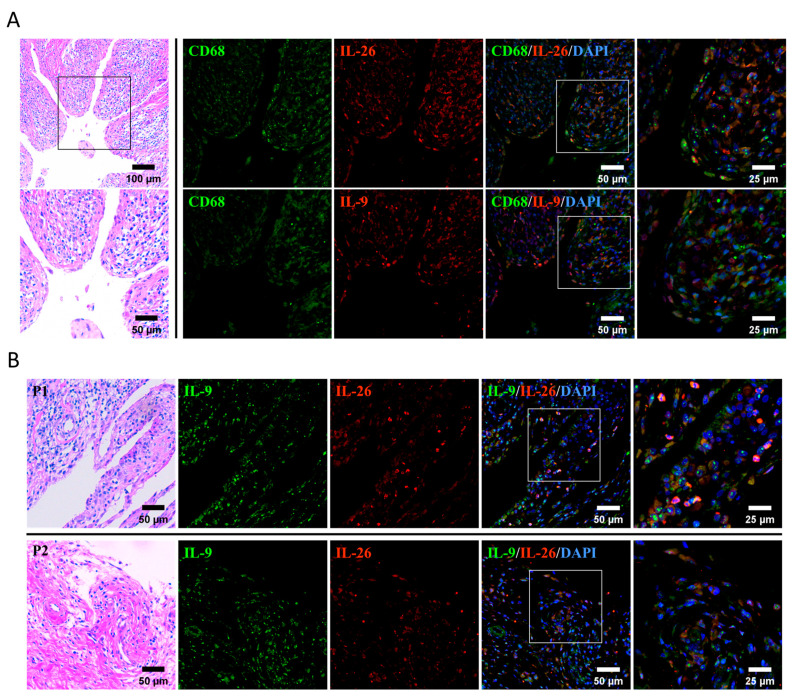
IL-26^+^ and IL-9^+^ macrophages were highly expressed in synovial tissue in RA. Synovial tissue from RA patients was stained by H&E or immunofluorescence. DAPI (blue) was used to mark the location of cell nuclei. (**A**) CD68 antibody (green) was used to identify macrophage cells and was counterstained with IL-9 or IL-26 antibody (red). (**B**) Counterstaining of IL-9 (green) and IL-26 (red) antibodies showed the overlapping sites of two cytokines in RA synovial tissue.

**Figure 2 ijms-24-07526-f002:**
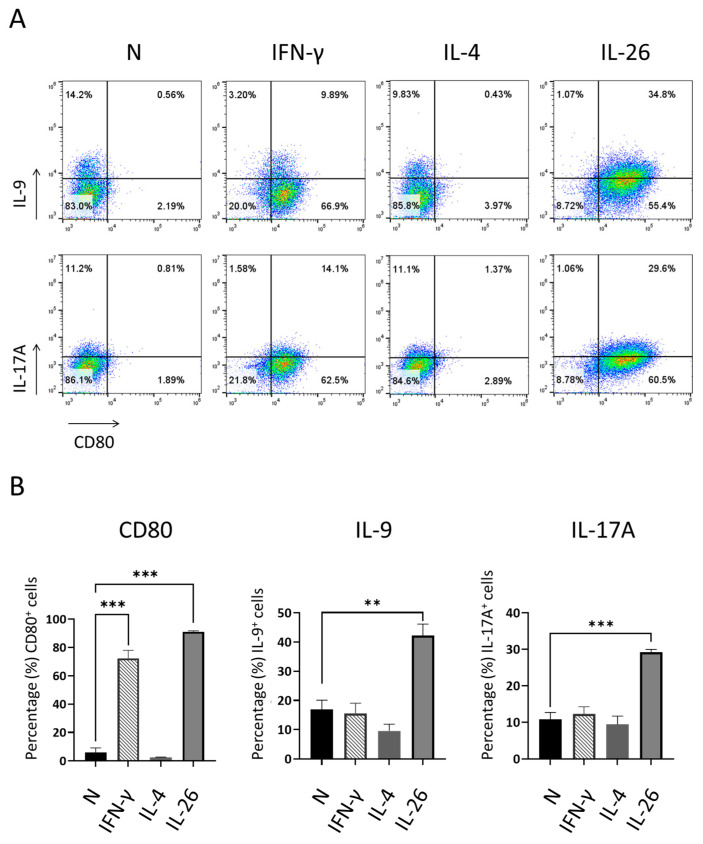
Effect of IL-26 on IL-9 cytokines expression and CD80 macrophage differentiation in RAW264.7. RAW264.7 cells were treated with IFN-γ (20 ng/mL), IL4 (20 ng/mL), or IL26 (60 ng/mL) for 18 h. After incubation, representative results (**A**) and summarized percentages of CD80, IL-9, and IL17A (**B**) were measured by flow cytometry. Results are the means ± SD of three independent experiments (*n* = 3) (** *p* < 0.01, *** *p* < 0.001, as multiple comparison significance between normal (N) control, IL-26, or IFN-γ group).

**Figure 3 ijms-24-07526-f003:**
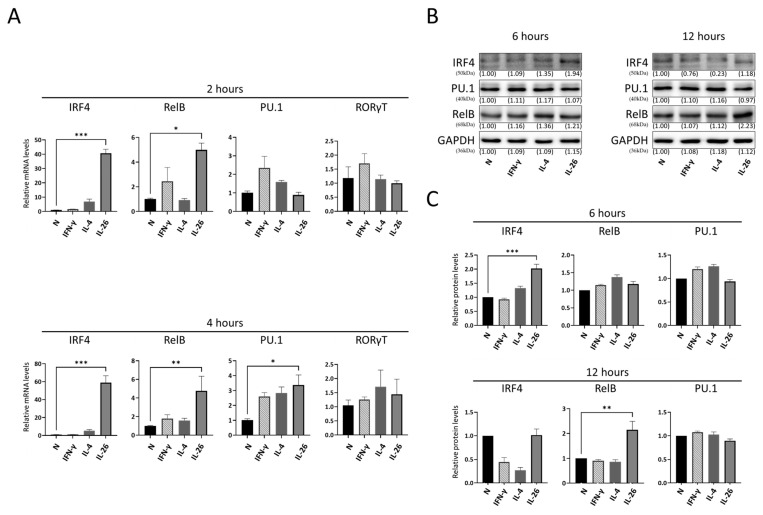
Effect of IL-26 on IL-9- and IL17A-related transcription factor expression in RAW264.7. RAW264.7 cells were treated with IFN-γ (20 ng/mL), IL-4 (20 ng/mL), or IL-26 (60 ng/mL). After 2 or 4 h incubation, mRNA gene expression level of IRF4, PU.1, RelB, and RORγt was measured by real-time PCR (**A**). After 6 or 12 h incubation, protein levels of IRF4, PU.1, and RelB were measured by Western blot (**B**,**C**). GAPDH was the loading control for checking equal amounts of cDNA or protein in each sample. Results are the means ± SD of three independent experiments (*n* = 3) and normalized to N group (* *p* < 0.05, ** *p* < 0.01, *** *p* < 0.001, as multiple comparison significance between each group).

**Figure 4 ijms-24-07526-f004:**
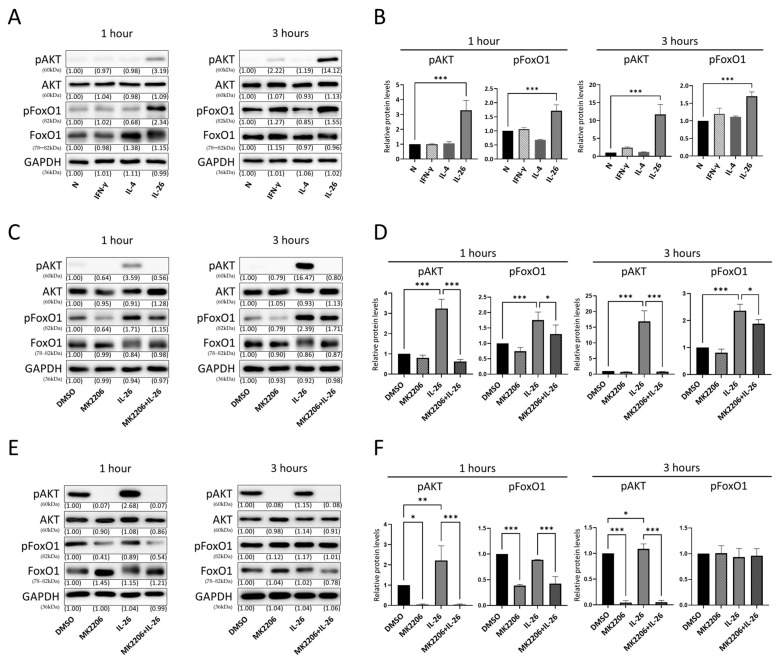
Effect of IL-26 on AKT and FoxO1 activation in RAW264.7 and THP-1. RAW264.7 cells were serum-starved for 12 h then treated with IFN-γ (20 ng/mL), IL-4 (20 ng/mL), or IL-26 (60 ng/mL) for 1 or 3 h to detect phosphorylated or nonphosphorylated AKT and FoxO1 protein by Western blot (**A**,**B**). RAW264.7 (**C**,**D**) and PMA-stimulated THP-1 cells (**E**,**F**) were serum-starved for 12 h and consecutively treated with p-Akt inhibitor (MK2206, 5 uM) or solvent (0.1% DMSO) for 1 h then concurrently treated with or without IL-26 for 1 or 3 h to detect phosphorylated or nonphosphorylated AKT and FoxO1 protein by Western blot. GAPDH was the loading control for checking equal amounts of protein in each lane. Results are the means ± SD of three independent experiments (*n =* 3) and normalized to N group (* *p <* 0.05, ** *p <* 0.01, *** *p <* 0.001, as multiple comparison significance between each group).

**Figure 5 ijms-24-07526-f005:**
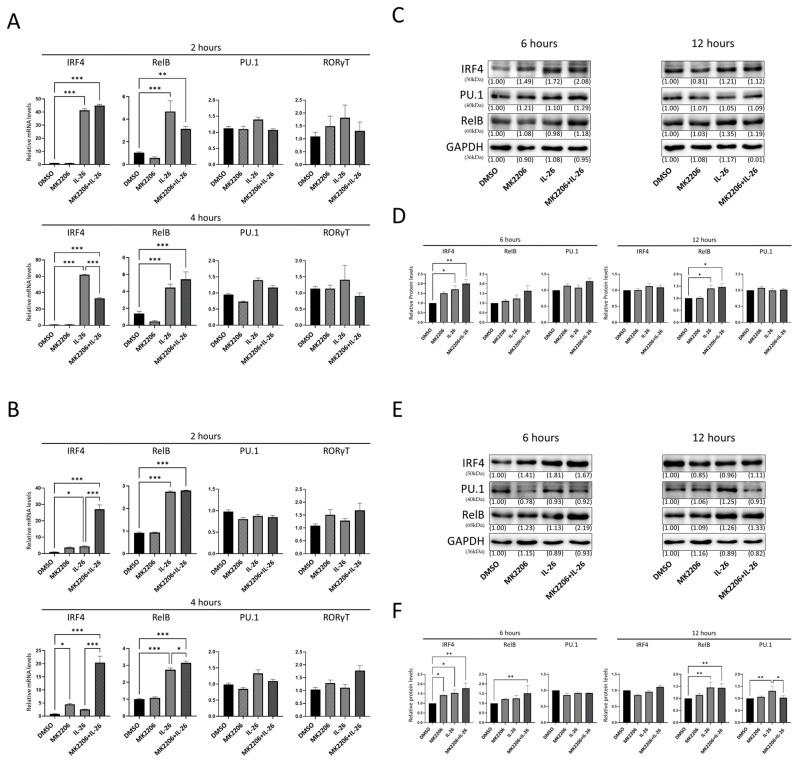
Effect of IL-26 plus MK2206 on IL-9 and IL17A regulated transcriptional gene expression in RAW264.7 and THP-1 cells. RAW264.7 (**A**,**C**,**D**) and PMA-stimulated THP-1 cells (**B**,**E**,**F**) were pretreated with p-Akt inhibitor (MK2206, 5 uM) or solvent (0.1% DMSO) for 1 h then concurrently treated with or without IL-26 (60 ng/mL). After 2 or 4 h incubation, gene expression levels of IRF4, PU.1, RelB, and RORγt were measured by real-time PCR (**A**,**B**). After 6 or 12 h incubation, protein levels of IRF4, PU.1, and RelB were measured by Western blot (**C**–**F**). GAPDH was the loading control for checking equal amounts of cDNA or protein in each sample. Results are the means ± SD of three independent experiments (*n =* 3) and normalized to N group (* *p <* 0.05, ** *p <* 0.01, *** *p <* 0.001, as multiple comparison significance between each group).

**Figure 6 ijms-24-07526-f006:**
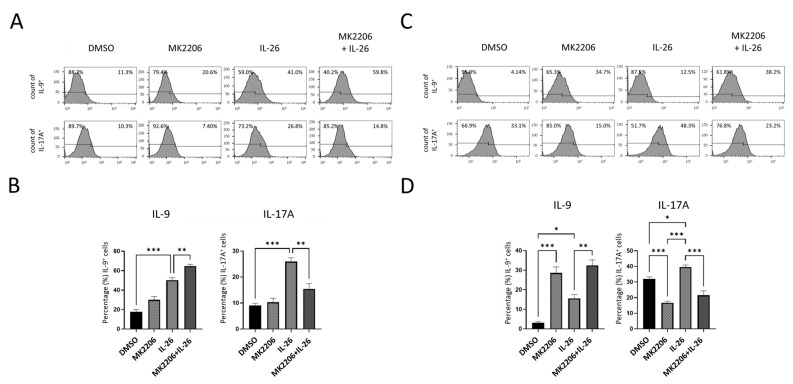
Effects of IL-26 plus MK2206 on IL-9 and IL-17A cytokines expression in RAW264.7 and THP-1. RAW264.7 (**A**,**B**) and PMA-stimulated THP-1 cells (**C**,**D**) were pretreated with p-Akt inhibitor (MK2206, 5 uM) or solvent (0.1% DMSO) for 1 h then concurrently treated with or without IL-26 (60 ng/mL) for 24 h. After incubation, representative results (**A**,**C**) and summarized percentages of IL-9, IL17A, and CD80 (**B**,**D**) were measured by flow cytometry. Results are the means ± SD of three independent experiments (*n =* 3) (* *p <* 0.05, ** *p <* 0.01, *** *p <* 0.001, as multiple comparison significance between each group).

**Figure 7 ijms-24-07526-f007:**
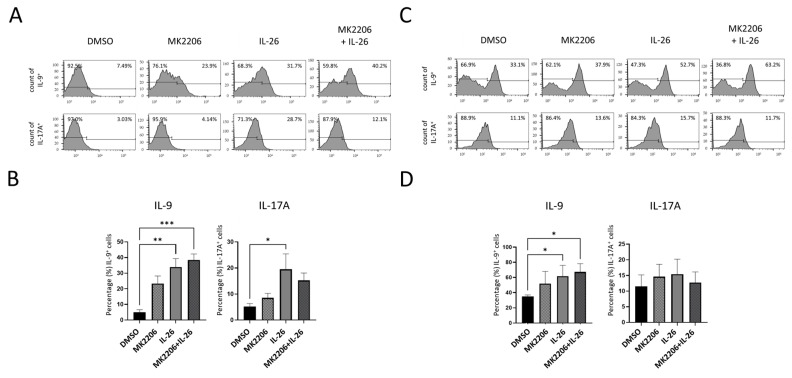
Effects of IL-26 plus MK2206 IL-9 and IL-17A cytokines expression in murine BMDM and human PBMC. BMDM (**A**,**B**) and PBMC cells (**C**,**D**) were pretreated with p-Akt inhibitor (MK2206, 5 uM) or solvent (0.1% DMSO) for 1 h then concurrently treated with or without IL-26 (60 ng/mL) for 24 h. After incubation, representative results (**A**,**C**) and summarized percentages of IL-9 and IL17A (**B**,**D**) were measured by flow cytometry. Results are the means ± SD of four independent experiments (*n =* 4) (* *p <* 0.05, ** *p <* 0.01, *** *p* < 0.001, as multiple comparison significance between each group).

## Data Availability

All data supporting the finding of this study are available within the article and its Appendix A.

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
