# Peer review of "Interleukin 26 Induces Macrophage IL-9 Expression in Rheumatoid Arthritis"

_ijms, 2023, doi:10.3390/ijms24087526_

Round 1

Reviewer 1 Report

Yi-Hsun Wang and colleagues performed a study where they demonstrated the effects on IL-26 on the modulation of IL-9 and IL-17 expression and the downstream pathways in macrophages. The results provide new insights usefully to understand the pathogenesis of rheumatoid arthritis and to formulate new therapeutic strategies, however there are some aspects that need to be clarified:

Minor concerns

a)     Introduction: this section should be improved 

b)     An overall signaling pathway of the effect of IL-26 would help readers to comprehend the hypothesis.

c)     The editorial office invites authors who performed western blot experiments to upload the full original images. The full original images provided by authors seem parts of the membranes and only one or two molecular weight indicators are reported, please clarify. 

e)    Methods: please provide more details, i.e real-time PCR, how the relative expression was calculated? 

Author Response

See my rely in attached file, many thanks.

Reviewer 2 Report

This study by Wang HY et al., provided data on IL-26 induced macrophage IL-9 and IL-17A expression via AKT-FoxO1 dependent manner.

While this study provide detailed investigations on the hypothesis, I would like to ask if authors have any convinving evidence from mouse models of RA? i.e. Targeting IL-26 in in vivo!

Minor:

Results; Line 65-IF is used to analyze Macrophages coexpression of IL-26 and IL-9. Authors should correct their statement that the purpose of H&E staining was for indetification of leukocyte infiltration in the synovium.  Figure representation is confusing in multiple ways; It appears like authors used same patient sample, but the panel for CD68/IL-26 presented in 200 µm, where as panel for CD68/IL-9 presented in 100µm, surprisingly that the scalebar length is similar. The bottom panel CD68/IL-9/DAPI square box marking was overlaping the scale bar, so please move the scalebar to the left for a better respresentation. CD68 staining does not look promising. Did authors consider using a negative control and isotype control for each panel?

Figure 1B: Square boxed area of the images are not matching in the far-right image. 

Author Response

(The authors gave the same response as above.)

Reviewer 3 Report

This study evaluates the role of IL-26 in regulation of IL-9 and IL-17 expression in rheumatoid arthritis. The authors show that stimulation of macrophages/monocytes with IL-26 directly induces the production of IL-9 and IL-17A via the AKT-FoxO1 pathway as well as differentiation to M1 macrophage type. This relation is the novelty of the manuscript. The authors confirmed their findings on mRNA and protein levels explaining the mechanisms of action and involved signaling pathways. The manuscript is well illustrated.

Areas to improve:

1) I recommend showing the percentages in the flow cytometry figures (Figures 2, 6 and 7) for easier comparison. Watching Figure 2B (IL-17A), the column for IL-4 is lower than N (less than 10%), but in Figure 2A (second row, IL-17A) can be seen that the positive population for IL-4 is higher than N?!

2) In Figures 3, 4, and 5, in the Y-axis is given relative ..... to what? Please, write this in the figures’ legend. How will be explained the statistical difference in the expression levels of IRF4 (Fig.3A, 4 h) and the equal protein levels of IRF4 (Fig.3C, 12 h)?

3) Please, give information in Materials and Methods about the culturing conditions for the cell lines and primary cultures used.

4) Line 220: Give the reference.

The English needs to be improved.

For example line 18 “…is FULL of proinflammatory cytokines”, line 39 “…SYNOVIAL joints”, line 42 “…synoviocyteS”, line 44 “…SIGNIFICANTLY detected”, line 57 “…many cell sources, like macrophages”, line 75 “…macrophage cellS”, line 81 the sentence is not completed “To test our hypothesis and get clear image that IL-9 was directly stimulated by IL-26 in macrophages…….WHAT IS DONE?”, lines 262-263 “…sections were BAKING at 65 degrees…”

Author Response

See my reply in attached file, many thanks.
